# Definition of Castrate Resistant Prostate Cancer: New Insights

**DOI:** 10.3390/biomedicines10030689

**Published:** 2022-03-17

**Authors:** Juan Morote, Adriana Aguilar, Jacques Planas, Enrique Trilla

**Affiliations:** 1Department of Urology, Vall d’Hebron Hospital, 08035 Barcelona, Spain; adrianaaguilargonzalez1995@gmail.com (A.A.); jplanas@vhebron.net (J.P.); etrilla@vhebron.net (E.T.); 2Department of Surgery, Universitat Autònoma de Barcelona, 08193 Barcelona, Spain

**Keywords:** prostate cancer, castration-resistance, PSMA-PET, testosterone, free-testosterone, luteinising-hormone

## Abstract

The term castrate resistant prostate cancer (CRPC) was initially proposed by the Prostate Cancer Working Group 2 in 2008 to define the state of clinical and/or biochemical progression of prostate cancer (PCa) in an environment with very low serum testosterone concentration. Clinical progression is based on the radiological imaging proposed by the Response Evaluation Criteria in Solid Tumors (RECIST) adapted to PCa. Biochemical progression is defined as an over 25% increase in serum prostate-specific antigen within two consecutive measurements separated by at least one week, and an absolute value above 2.0 ng/mL. Finally, the castrate environment is usually defined as a serum testosterone concentration maintained below 50 ng/dL or 1.7 nmol/dL. This definition does not incorporate the new and more accurate imaging modalities to assess clinical progression and the capability of the new biochemical measurements to assess the true castration environment. Ga-68-PSMA-11 PET CT/MRI and whole-body MRI are the new imaging modalities that should replace the classic thoracic CT scan, abdomino-pelvic CT scan, and technetium 99-m bone scintigraphy. In addition, Ga-68-PSMA-11 PET is the current basis for the new therapies targeting metastatic sites. Moreover, the current methods for measuring the very low serum testosterone concentrations in clinical laboratories are the widespread chemiluminescent assays, which are inappropriate, while LC-MSMS is the only method recommended to assess the castrate environment. In addition, recent research shows that serum luteinising hormone concentration associates better than serum testosterone with the castration environment, even when it is measured with LC-MSMS. In summary, the current definition of CRPC seems outdated. An extensive update to diagnose true CRPC is also needed to differentiate CRPC men with M0 (non-metastatic) from those with M1 (metastatic) CRPC. WC: 277.

## 1. Introduction

The hormone dependence of prostate cancer (PCa) was discovered by the urologist Charles Brenton Huggins in 1941 [1]. Huggins was awarded the Nobel Prize in Physiology and Medicine in 1966 for his contribution to the treatment of PCa with what is currently named androgen deprivation therapy (ADT) [2]. Huggins and Hodges introduced surgical castration and the administration of oestrogens to decrease the serum testosterone levels in men with advanced PCa [1]. Between 1960 and 1975, the Veterans’ Administration Cooperative Urological Research Group (VACURG) conducted three major randomised clinical trials comparing various endocrine treatments for patients newly diagnosed with advanced PCa. The main conclusions regarding hormonal treatment that emerged from these studies were: (1) increased hazard of cardiovascular death after therapy with 5 mg diethylstilbestrol (DES); (2) orchiectomy plus DES is no better than orchiectomy or DES alone; (3) the effects of 1.0 and 5.0 mg DES on cancer are equivalent; (4) reduced cardiovascular hazard from therapy with 1.0 mg DES [3].

Another significant contribution to the endocrine treatment of PCa came from the endocrinologists Roger Guillemin and Andrew Victor Schally, who contributed to the understanding of the hypothalamic–pituitary axis and the discovery of the chemical structure of peptide hormones, as well as the gonadotrophin releasing hormones (GnRH) [4]. These discoveries were also awarded the Nobel Prize in Physiology and Medicine in 1977 and prompted the development of the current luteinising hormone-releasing hormone (LH-RH) agonists. Buserelin was the first LH-RH agonist introduced to treat advanced PCa outside the US in 1985 [5], and goserelin was the first Food and Drug Administration (FDA)-approved LH-RH agonist in 1989 [6]. Therefore, medical castration was introduced to facilitate reversible ADT and avoid the secondary psychological effects of bilateral orchiectomy [7]. 

Castration resistance is defined as the progression of disease in a castration environment, and it precedes hormone resistance, which is defined as the progression of disease despite whichever hormonal manipulation is added to castration. The term castrate-resistant prostate cancer (CRPC) was proposed by the Prostate Cancer Working Group 2 (PCWG2) in 2008 to describe clinical and/or biochemical progression in a castration environment [8]. The clinical progression of solid tumours is usually based on the Response Evaluation Criteria in Solid Tumors (RECIST 1.1), in which the criterion for progression is an increase of at least 20% in the longest diameter of target lesions, taking as reference the smallest longest diameter recorded since the treatment started or the appearance of one or more new lesions [9]. The current specific criteria to define the progression of PCa after castration are those proposed by the PCWG 2, in which biochemical progression is defined as an over 25% increase in serum prostate-specific antigen (PSA) within two consecutive measurements separated by at least one week, with a 2.0 ng/mL minimum increase over the starting value, and PSA doubling time is also incorporated to predict the aggressiveness of progression [10]. For RECIST 1.1 progression of visceral metastases, at least a 2 cm length of lymph nodes on computed tomography (CT) scan or magnetic resonance imaging (MRI) is required. Prostate and/or prostate bed progression can be assessed by CT scan, pelvic MRI, endorectal MRI, transrectal ultrasound, or incidental positron emission tomography (PET)/CT. Bone progression is defined as two or more new lesions appearing in technetium-99 m scintigraphy, although confirmation by CT scan or MRI is required when results are ambiguous. Finally, the castrate environment is defined as a serum testosterone concentration below 50 ng/dL or 1.7 nmol/dL [8]. The current EAU (European Association of Urology) CRPC definition is based on the biochemical and/or clinical progression according to PCWG 2 and RECIST 1.1 criteria being the serum testosterone below 50 ng/mL, Figure 1 [11]. 

We aim to review the new insights from more efficient imaging modalities than those currently proposed by RECIST 1.1 for the assessment of PCa clinical progression, as well as the new insights regarding the best methods to assess the castration environment, in order to improve the current definition of CRPC and the proper selection of patients for treatment.

## 2. Effects of Castration on Serum Hormones

Castration reduces the amount of circulating testosterone produced in the Leydig cells of the testis under the stimulation of serum luteinising hormone (LH) liberated from the pituitary gland, which accounts for 95% of circulating testosterone. The remaining 5% of circulating testosterone is produced in the adrenal cortex under the stimulation of the adrenocorticotropic hormone. A significant difference in the concentration of serum LH is observed after medical vs. surgical castration: while a drastic decrease in serum LH is observed after LH-RH agonist or antagonist administration, the serum LH remains high after surgical castration. The main clinical difference of the new LH-RH antagonist is their immediate block of pituitary LH-RH receptors which is translated in a rapid decrease of serum testosterone to castrate levels in less than 24 h over the classic agonist which induced this decrease in one month. In one study conducted at 150 sites in the US and Mexico that included 1191 men with PCa undergoing GnRH agonist treatment and 59 subjected to bilateral orchidectomy, the mean serum levels of LH (range) were 1 UI/L (1–526) and 159 UI/L (33–369), respectively [12]. Testosterone is the main sex hormone in men, regulating many functions such as libido, bone mineral density, red blood cell count, male characteristics, and male behaviours. Total blood levels of testosterone comprise bound and free forms. Most of the circulating testosterone is bound to either albumin or sex hormone-binding globulin (SHBG), which is a protein produced by the liver. To protect from testosterone degradation, SHBG serves to regulate the amount of free testosterone that is available for biological activity by keeping it bound and therefore inactive. Free testosterone is the metabolically active form of testosterone that carries out the biological functions associated with its activity [13]. An individual’s free testosterone is internalised into the cells after the cytoplasmatic 5-α-reductase enzyme transforms free testosterone to 5-dihydrotestosterone, which is able to promote androgen receptor dimerisation and then translocation into the nucleus, where it makes possible deoxyribonucleic acid replication [14].

## 3. Methods for Measuring Total Serum Testosterone

Castration results in a significant reduction of serum testosterone levels, and sensitive assays are required to accurately measure these low concentrations. Although several studies have validated techniques for the measurement of relatively high levels of serum testosterone (above 100 ng/dL) [15], the castrate levels of testosterone fall much lower [16]. 

The FDA has never defined the castrate level of serum testosterone. However, many documents support the use of a serum testosterone level below 50 ng/dL as a standard for FDA approval of castration products [17]. These data came from the orchiectomy arms of the VACURG trials [3], and 50 ng/dL was the lowest limit of detection of the radioimmunoassays (RIAs) used at that time [18]. The European Medical Association’s Sixth International Consultation on New Developments in Prostate Cancer and Diseases agreed that because total serum testosterone levels below 20 ng/dL (measured with chemiluminescent assay [CLIA]) are typical in men who have undergone bilateral orchiectomy, this threshold should be utilised for chemical castration [13]. However, the American Urological Association lists 50 ng/dL as the threshold for chemical castration [19], and the European Association of Urology PCa guidelines also establish the threshold of chemical castration at 50 ng/dL, although they noted that testosterone levels below 20 ng/dL are associated with an improvement in outcomes compared with levels within the 20–50 ng/dL range [11].

The double-isotope derivative dilution technique made possible the initial measurements of serum testosterone in the late 1960s [20]. RIAs were used, starting in the 1970s, but they have limited accuracy and sensitivity to measure low concentrations [17]. Measurement of the true spread of serum testosterone in clinical laboratories was enabled by the sensitive and automatable CLIAs, with the Chiron/Ciba-Corning Diagnostics ACS: 180 (Norwood, MA, USA) being the first immunoassay analyser approved for clinical use [21]. This technique permitted the assessment of low testosterone concentrations while also decreasing the waiting time for results. Oefelein et al. redefined the castrate level of serum testosterone with a CLIA as 20 ng/dL [16], and microelevations of serum testosterone were described in men undergoing GnRH agonist treatment [22,23]. In 2007, our group first demonstrated that lower serum testosterone levels were associated with longer survival free of castrate resistance, especially in men without microelevations over 32 ng/dL [24]. Other studies have also suggested that lower testosterone levels are associated with better follow-up [25]. All the studies analysing the influence of microelevations of serum testosterone on the findings during follow-up have been carried out with CLIAs [26]; in contrast, the studies presented to the FDA for the approval of agents for castration always used a type of chromatography before specific RIAs [27] and, more recently, liquid chromatography and tandem mass spectrometry (LC-MSMS), which is currently considered the gold standard method for testosterone testing, especially for low testosterone concentrations [28,29]. While the microelevations of serum testosterone over 50 ng/dL ranged between 3% and 12% in studies that used CLIAs for testosterone measurement, they ranged from 0% to 1% in the clinical trials in which LC-MSMS was used [23]. 

Between 2003 and 2004, two important studies compared the standard method of liquid or gas chromatography (GC) mass spectrometry with the available immunoassays (RIAs and CLIAs) in women and children [30] and adult men [31]. These studies concluded that RIAs and CLIAs usually overestimate low concentrations of serum compared to the standard LC/GC-MS measurements. In 2007, the International Society of Endocrinology stated that LC-MSMS is the recommended method to measure serum testosterone in children and women [15]. In one study analysing the serum testosterone levels of 249 men with PCa undergoing GnRH agonist treatment, we observed no correlation between the levels measured with two commercial CLIAs, as the rate of microelevations over 50 ng/dL ranged between 21.3% and 0.8% [32]. In a more recent study, we compared the serum testosterone levels of 126 PCa patients undergoing continuous GnRH agonist treatment using CLIA and LC-MSMS. The median serum testosterone was 14.0 ng/dL (range 2.0–67.0) when measured with LC-MSMS and 31.9 ng/dL (range 10.0–91.6) with CLIA (*p* < 0.001). The serum testosterone levels measured with LC-MSMS were below 20 ng/dL in 65.9%, between 20 and 50 ng/dL in 31.7%, and over 50 ng/dL in 2.4% of patients. These rates were 27, 57.1, and 15.9%, respectively, when testosterone was measured with CLIA (*p* < 0.001). The castrate levels of serum testosterone measured with LC-MSMS and CLIA were 39.8 ng/dL (95% CI 37.1–43.4) and 66.5 ng/dL (95% CI 62.3–71.2), respectively. We concluded that CLIA overestimated the serum testosterone levels in PCa patients undergoing LH-RH agonist therapy. More than 15% of CLIA measurements were over 50 ng/dL, while 2.4% of LC-MSMS measurements were above this value. The estimated castrate level of serum testosterone from the appropriate method of measurement was lower than the currently used up to 50 ng/dL [33].

## 4. Methods for Evaluating the Castration Environment

There are at least three methods for evaluating the castration environment. 

### 4.1. Total Serum Testosterone

The classical method for evaluating the castration environment in patients with PCa is the assessment of total testosterone in serum [17]. Serum testosterone testing was initially used to assess the effectiveness of DES as a method of castration compared to surgical castration [3]. The castrate level of serum testosterone was established as 50 ng/dL, which was the lowest level detected by the RIA used in the 1960s [20]. Measuring the true spread of serum testosterone was possible after the introduction of automatable CLIAs [21]. However, some research demonstrated that immunoassays usually overestimated low levels of serum testosterone and had low reproducibility [30,31]. This evidence led the International Society of Endocrinology to recommend against using immunoassays for the measurement of testosterone levels in children and women, which should be measured with the classic mass spectrometry following liquid or gas chromatography [15]. This issue for serum testosterone testing in men with PCa undergoing castration has recently been investigated, and a similar recommendation has been proposed [32,33]. Today, LC-MSMS is completely automated and offers an accurate, reproducible, and quick measurement of serum testosterone, with the high price being the only disadvantage compared to CLIAs [34]. 

### 4.2. Free Serum Testosterone

Testing of free serum testosterone should be the ideal method for evaluating the castration environment because it is the active form of testosterone. Less than 5% of the total amount of serum testosterone is free, and it can diffuse into the cells and bind to the androgen receptor after being converted to dihydrotestosterone by the 5-α-reductase enzyme [35]. In addition to its direct measurement in serum or plasma, the free testosterone value can also be calculated using the Vermeulen method, a formula relying on the total testosterone, SHBG, and albumin concentrations measured from immunoassay [36]. In patients with advanced PCa, being subjected to castration and achieving lower free serum testosterone seems to be associated with better overall survival [17]. In 2018, Schweizer et al. analysed the effects of different ADTs on free serum testosterone levels. Using RIA to measure free serum testosterone, the authors did not find a difference in levels when comparing patients subjected to surgical vs. medical castration [12]. In 2005, we analysed free serum testosterone levels in 135 patients with advanced PCa undergoing continuous luteinising hormone-releasing hormone (LH-RH) analogue treatment. After establishing cutoffs for castration levels of total testosterone below 50 ng/dL and for free testosterone below 1.7 pg/mL, 86% of patients met the cutoff for total testosterone after treatment, while 95% met the cutoff for free testosterone. Although correlation was observed between the testosterone measurements, the authors concluded that total and free testosterone may report complementary information [37]. One study analysed the relationships between serum total testosterone, SHBG, and the calculated free testosterone in PCa patients who underwent surgical castration or oestrogen administration. The third of the 33 patients subjected to orchiectomy who had the lowest free testosterone or total testosterone levels exhibited a better survival over 2 years than the two-thirds who had higher levels. Despite these findings, there was no evidence of an increase in free testosterone level accompanying the clinical progression in these patients. In addition, free testosterone was lower in oestrogen-treated patients than in orchiectomised patients [38]. In 2017, we analysed free and total serum testosterone in 29 patients with advanced PCa undergoing continuous LH-RH analogue treatment. The purpose of the study was to compare serum free and total testosterone levels to predict survival free of castration resistance. Total testosterone cutoffs of 50, 32, and 20 ng/dL were established, and free testosterone cutoffs were set at 1.7, 1.1, and 0.7 pg/mL. The lowest threshold that detected a significant difference in survival free of castration resistance was 1.7 pg/mL of free testosterone. Therefore, free serum testosterone was a better predictor than total serum testosterone in predicting castration resistance [39]. 

Since the International Society of Endocrinology position statements regarding total testosterone testing in 2007 [15] and 2010 [40], no position statement regarding free testosterone testing has been published. The Vermeulen method to calculate the free testosterone serum concentration was designed using immunoassay measurements of total serum testosterone [36], and no validation of this method has been performed with castrate levels of testosterone and other measurement methods. In addition, no evidence exists regarding the sensitivity or reproducibility of free serum testosterone measurements, whether calculated or directly measured with available immunoassays. 

### 4.3. Serum Luteinising Hormone

The hypothesis that measurement of serum LH could be used as a method to assess the castrate environment was established from the observation that patients with PCa who underwent continuous medical castration with LH-RH analogues exhibited serum testosterone levels over 50 ng/dL and low serum LH concentrations. We believe that, in this scenario, extra-testicular production of testosterone or false microelevation of serum testosterone is possible [12]. In addition, by measuring serum LH, Garnick and Mottet have efficiently monitored the switch from LH-RH antagonist to LH-RH analogues in PCa patients [41].

Serum LH and total serum testosterone were measured by CLIAs in 1091 men, 488 PCa patients undergoing treatment with LH-RH analogues, referred to as “on LH-RH agonists”; 303 PCa patients in whom LH-RH analogue was withdrawn (“off LH-RH agonists”); and 350 men with suspicion of PCa who never received LH-RH analogues (“no LH-RH analogues”). In addition, in a validation cohort of 147 PCa patients whose total serum testosterone was measured by LC-MSMS, 124 were “on LH-RH analogues” and 19 were “off LH-RH analogues”. The area under the curve (AUC) for distinguishing patients “on LH-RH agonist” from those “off LH-RH analogues” was 0.997 for serum LH and 0.740 for total serum testosterone measured by CLIA (*p* < 0.001). The castrate threshold of serum LH was established as 1.1 UI/L. The AUCs of serum LH, serum total testosterone measured by CLIA, and serum total testosterone measured by LC-MSMS in the validation cohort were 1.000, 0.646, and 0.814, respectively (*p* < 0.001). The accuracy of distinguishing PCa patients “on LH-RH analogues” from those “off LH-RH analogues”, using the thresholds of 1.1 U/L for serum LH and 50 ng/dL for serum total testosterone measured by CLIA and LC-MSMS, were 98.6, 78.3, and 89.5%, respectively (*p* < 0.001). This study concluded that regardless of the method used to measure serum testosterone, serum LH was more efficient in assessing the castrate environment, as it efficiently distinguished between patients “on LH-RH agonist” and those “off LH-RH agonists” [42].

We have recently analysed whether serum LH could distinguish between optimal and suboptimal castration in 136 patients with PCa undergoing continuous LH-RH analogue treatment. For this purpose, optimal castration was defined as a serum total testosterone level below 20 ng/dL (measured by CLIA and LC-MSMS), and the serum LH level ranges were < 0.12 UI/L and 0.13–1.1 UI/L. In patients with optimal castration, the rate of LC-MSMS serum testosterone levels below 20 ng/dL was 78.3%, while the rate was 21.7% in patients with suboptimal castration (*p* < 0.001). The rates of CLIA serum measurements below 20 ng/dL were similar in the optimal and suboptimal castration groups (53.6 vs. 46.4%, *p* = 295). We conclude that serum LH was significantly associated with serum testosterone (when it was appropriately measured) in PCa patients undergoing continuous medical castration. A serum LH level of 0.12 U/L was associated with optimal castration, defined as serum testosterone < 20 ng/dL [43]. 

Future studies verifying that optimal castration based on serum LH measurement is associated with better follow-up than patients with suboptimal castration and those with no castrate environment are needed. 

## 5. Imaging Modalities for the Diagnosis of Castration-Resistant Prostate Cancer

Beyond the current imaging modalities proposed by the RECIST guidelines [9], which mainly comprise CT scans and MRI, the PCWG2 also considered technetium-99 m bone scintigraphy to assess the progression of bone metastases when more than two hot spots appear; however, CT scans or MRI are required when results are ambiguous [8]. 

PET-whole body CT/MRI and whole body multiparametric MRI are new imaging modalities improving the classic CT scan and MRI for the detection of new metastatic sites and the bi-dimensional assessment of progression using different radiotracers as Ga-68-PSMA-11 which is one of the most specific for prostate cancer cells [44].

Currently, there is solid evidence on the re-staging efficacy of Ga-68-PSMA11-PET in men with non-metastatic CRPC [45,46]. Frendler et al. conducted a multicenter and retrospective study of 200 patients with CRPC with serum PSA levels over 2 ng/mL, a PSA doubling time below 10 months, and/or a Gleason score ≥ 8 in whom conventional imaging showed the absence of metastasis. The Ga-68-PSMA11-PET/CT showed positive findings in 98% of the patients. In 44% of the patients, positive results were observed in the pelvis, with 24% of patients having progression or recurrence in the prostate or prostate bed. Furthermore, in 55% of the patients who developed disease recurrence, metastases were found in the extra-pelvic lymph nodes (39%), bone (24%), and visceral organs (6%) [47]. Using Ga-68-PSMA11-PET/CT, Fourquet et al. analysed 30 non-metastatic PCa patients with increasing serum PSA after medical castration and observed positive findings in all 20 patients whose serum PSA was over 2 ng/mL and in 7 of 10 patients whose serum PSA stayed below 2 ng/mL. In the 7% of cases with positive PSMA-PET, lesions were confined to the prostate gland or prostate bed. In addition, 20% of the patients had oligometastatic disease with fewer than three lesions and 63% had polymetastatic disease [48].

Whole-body MRI and localised multiparametric MRI are other candidate imaging modalities for re-staging patients with CRPC [49]. Head-to-head comparative studies between Ga-68-PSMA11-PET/CT/MRI and whole-body MRI are difficult; however, it seems that the efficiency of both imaging modalities depends on the metastatic sites. In any case, both imaging modalities are more efficient than those that are currently recommended [50]. 

Thus, the current definition of CRPC based on the classic imaging modalities does not represent the true stage of the disease [51]. Additionally, new focal treatments for metastatic CRPC based on PET imaging and adding certain conjugated drugs with biological activity against prostate cells are under development [44,52,53]. Figure 2 gather up the proposed components of a new definition of castration resistant prostate cancer.

## 6. Conclusions

The current definition of CRPC is outdated. The new insights in the assessment of the castration environment and the true re-staging with novel imaging modalities will improve the definition and classification of patients with CRPC. Moreover, novel focalised treatments for metastatic disease based on some of the new imaging modalities are currently being developed.

## Figures and Tables

**Figure 1 biomedicines-10-00689-f001:**
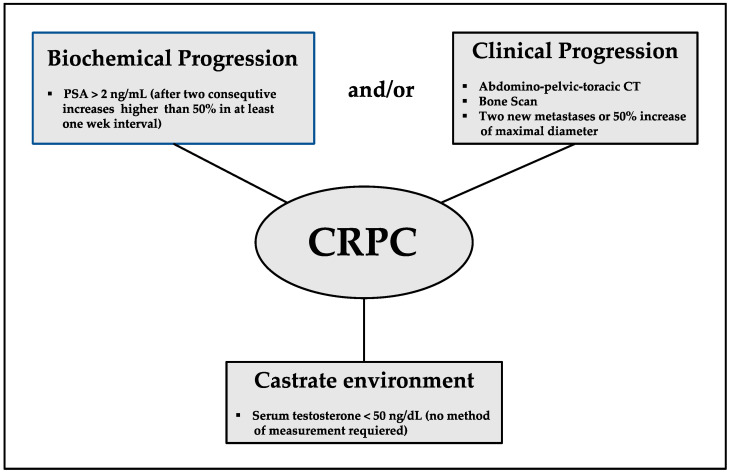
Current definition of Castrate Resistant Prostate Cancer.

**Figure 2 biomedicines-10-00689-f002:**
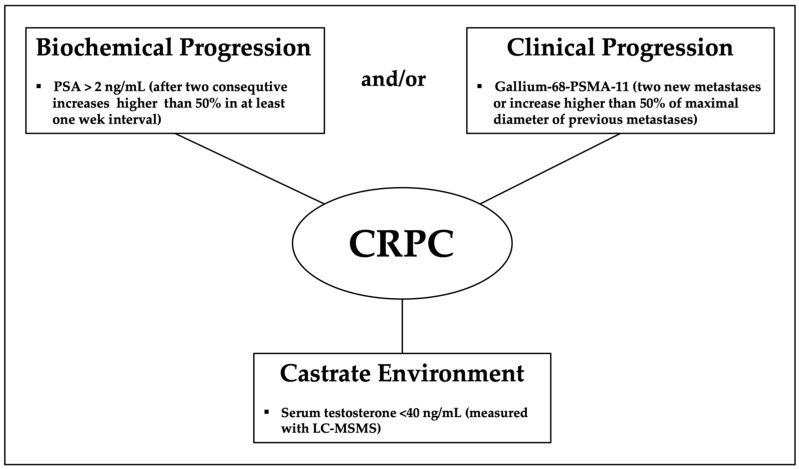
Proposed definition of Castrate Resistant Prostate Cancer.

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
