# Peer review of "Definition of Castrate Resistant Prostate Cancer: New Insights"

_biomedicines, 2022, doi:10.3390/biomedicines10030689_

Round 1

Reviewer 1 Report

This review article collates new insights into the parameters that define castrate resistant prostate cancer (CRPC) and contend that the current working definition requires updating.

The authors have forensically mined and appraised the Literature underpinning the definition of CRPC, its diagnosis, staging and patient management. The review focusses on evaluating the relative importance of key hormonal biomarkers and compares classical and emerging methodologies for determination of their levels, notably in the castration microenvironment.

The informative and balanced discussion of the merits and limitations of current tumor imaging techniques compared to data derived from the growing use of modern and more powerful imaging methods, in the clinical setting, presents an evidence-based argument for adoption of the latter as the preferred methods to more accurately measure serum biomarkers. 

Whilst acknowledging and identifying where further work is required regarding validation of modern imaging methods and their development, the evidence for their adoption provides a compelling case to uphold the contention that the current definition of CRPC is outdated and merits redefinition in the light of powerful imaging and improved limits of detection of biomarkers, in favor of improved treatment regimens and survival for patients. 

_______________________________________________________________________________

Recommendation: to capture a wider group of readers, even though abbreviations are defined in the text when they first appear, the manuscript would benefit from a table or list of abbreviations for quick and easy reference.

Minor point: 

The sentence: 'The castration level of serum testosterone using proper was below 50 ng/dL.' [Lines 170-171] should be revisited. There is something clearly missing from the sentence. Correction is required to clarify the point being made, to restore grammatical sense.     

Author Response

Agradecemos los comentarios y sugerencias del revisor.

  1. Hemos agregado en el manuscrito una tabla con una lista de siglas. Líneas 48-80
  2. Hemos cambiado la oración en las líneas 170-171 (ahora 206-7): El nivel de castración estimado de testosterona sérica del método de medición apropiado fue más bajo que el que se usa actualmente hasta 50 ng/dL 

Reviewer 2 Report

Dear authors, I read carefully your paper. The article can offer to the reader an interesting review on CRPC and is well written, but I would like to suggest you some improvements that can increase the general quality of the paper. Please look at the following suggestions:

PARAG.1: I believe that the historical part on ADT can be summarized while you can add something else on CRPC (for example what EAU guidelines suggest). Please at the end of the paragraph add the aim of the paper.

PARAG.2: please explain the difference between LHRH agonist and antagonist

PARAG.4: please elucidate better Vermeulen formula, even including albumin role.

PARAG.5: PSMA-PET is not the only other method to evaluate bone metastasis. Please include even other kinds of PET and their role.

Author Response

We thanks the comments and suggestions of the reviewer:

  1. Paragraph 1. The next paragraph in line 44-50 has ben suppressed: 5) Premarin (conjugated oestrogens of equine origin) and Provera (medroxyprogesterone acetate) are no better than 1.0 mg DES at the doses studied; and 6) decisions about hormone treatment at diagnosis depend on patient characteristics, mainly age and Gleason grade.

    The EAU definition of CRPC has been added in lines 113-116: The current EAU (European Association of Urology) CRPC definition is based on the biochemical and/or clinical progression according to PCWG 2 and RECIST 1.1 criteria being the serum testosterone below 50 ng/mL [19].

    The last sentence of the paragraph (lines 117-121) has been change to emphasise the objective of the study. We aim to review the new insights from more efficient imaging modalities than those currently proposed by RECIST 1.1 for the assessment of PCa clinical progression, as well as the new insights regarding the best methods to assess the castration environment, in order to improve the current definition of CRPC and the proper selection of patients for treatment.

  2. Paragraph 2. The next paragraph has been introduced en lines 130-133. The main clinical difference of the new LH-RH antagonist is their immediate block of pituitary LHRH receptors which is translated in a rapid decrease of serum testosterone to castrate levels in less than 24 hours over the classic agonist which induced this decrease in one month. 
  3. Paragraph 4. The sentence in lines 237-239 has been modify: In addition to its direct measurement in serum or plasma, the free testosterone value can be also calculated using the Vermeulen method, a formula relying on the total testosterone, SHBG, and albumin concentrations measured from immunoassay [36].
  4. Paragraph 5. The sentence in lines 322-325 has been modify. PET-whole body CT/MRI and whole body multiparametric MRI are new imaging modalities improving the classic CT scan and MRI for the detection of new metastatic sites and the bi-dimensional assessment of progression using different radiotracers as Ga-68-PSMA-11 which is one of the most specific for prostate cancer cell [44].Regarding the last function of PET, the sentence in lines 347-350 has been changed. Additionally, new focal treatments for metastatic CRPC based on PET imaging and adding certain conjugated drugs with biological activity against prostate cells are under development [44,52,53]

Reviewer 3 Report

This concise, well-conceived review presents summaries of current practices, including their efficiencies and limitations, in reference to the previously established guidelines for evaluation of castrate resistant prostate cancer. The topics discussed in the review are appropriate and cover all relevant clinical and laboratory measurements associated with this aspect of medicine. The level of detail is also appropriate for the types of readers who are likely to access the manuscript. Inclusion of salient historical milestones is similarly well-placed and informative for readers who are new to this area. English language is excellent and referencing is also very good. I do not feel that there are any major topics that have been omitted, especially as the manuscript is formatted as an ‘opinion piece’. As such, I feel that the manuscript is suitable for publication in its present form, following amendment of the following minor corrections.

  1. Line 50: GnRH stands for gonadotrophin releasing hormone (not gonadotrophin luteinising hormone).
  2. Line 107: ‘translocate’ should be ‘translocation’
  3. Gonadotrophin releasing hormone (GnRH) and luteinising hormone releasing hormone (LHRH) are two names for the same hormone, however, the authors have used both names in different places in their manuscript without explaining that they are the same molecule. My suggestion would be to choose one of the abbreviations for use throughout the manuscript. When it is first mentioned, its alternate name should be stated along with a short explanation or footnote describing which one will be used in the manuscript.

Author Response

We thanks the comments and suggestions of the reviewer.

  1. This sentence has been modified (lines 81-90). Another significant contribution to the endocrine treatment of PCa came from the endocrinologists Roger Guillemin and Andrew Victor Schally, who contributed to the understanding of the hypothalamic-pituitary axis and the discovery of the chemical structure of peptide hormones, as well as the gonadotrophin releasing hormones (GnRH) [4]. These discoveries were also awarded the Nobel Prize in Physiology and Medicine in 1977 and prompted the development of the current luteinising hormone-releasing hormone (LH-RH) agonists. Buserelin was the first LH-RH agonist introduced to treat advanced PCa outside the US in 1985 [5], and goserelin was the first Food and Drug Administration (FDA)-approved LH-RH agonist in 1989 [6]. Therefore, medical castration was introduced to facilitate reversible ADT and avoid the secondary psychological effects of bilateral orchiectomy [7].
  2. Translocate has been substitute by translocation (now in line 147).
  3. GnRH has been introduced in line as gonadotrophin releasing-hormone  (line 84) and thereafter LH-RH as de luteinising hormone-releasing hormone has now been used in the rest of manuscript.

Round 2

Reviewer 2 Report

No further requirements